# Renal Impairment Detectors: IGFBP-7 and NGAL as Tubular Injury Markers in Multiple Myeloma Patients

**DOI:** 10.3390/medicina57121348

**Published:** 2021-12-10

**Authors:** Karolina Woziwodzka, Jolanta Małyszko, Ewa Koc-Żórawska, Marcin Żórawski, Paulina Dumnicka, Artur Jurczyszyn, Krzysztof Batko, Paulina Mazur, Małgorzata Banaszkiewicz, Marcin Krzanowski, Paulina Gołasa, Jacek A. Małyszko, Ryszard Drożdż, Katarzyna Krzanowska

**Affiliations:** 1Department of Nephrology, Jagiellonian University Medical College, 30-688 Kraków, Poland; woziwodzka.karolina@gmail.com (K.W.); batko.krzysztof@gmail.com (K.B.); mbanaszkiewicz92@gmail.com (M.B.); mkrzanowski@op.pl (M.K.); paulinagolasa1@gmail.com (P.G.); 2Department of Nephrology, Dialysis and Internal Medicine, Medical University of Warsaw, 02-091 Warsaw, Poland; jolmal@poczta.onet.pl; 3Second Department of Nephrology and Hypertension with Dialysis Unit, Medical University of Bialystok, 15-276 Bialystok, Poland; ewakoczorawska@wp.pl; 4Department of Clinical Medicine, Medical University of Bialystok, 15-254 Bialystok, Poland; mzorawski@wp.pl; 5Department of Medical Diagnostics, Jagiellonian University Medical College, 30-688 Kraków, Poland; paulina.dumnicka@uj.edu.pl (P.D.); paulina.pater@uj.edu.pl (P.M.); ryszard.drozdz@uj.edu.pl (R.D.); 6Departament of Hematology, Jagiellonian University Medical College, Kopernika 17, 30-501 Kraków, Poland; mmjurczy@cyf-kr.edu.pl; 7First Department of Nephrology and Transplantology with Dialysis Unit, Medical University of Bialystok, 15-540 Bialystok, Poland; jacek.malyszko@umb.edu.pl

**Keywords:** multiple myeloma, urine insulin-like growth factor-binding protein 7, tissue inhibitor of matrix metalloproteinase 2, neutrophil gelatinase-associated lipocalin, tubular kidney injury, free light chains, biomarker

## Abstract

*Background and Objectives*: Urine insulin-like growth factor-binding protein 7 (IGFBP-7), tissue inhibitor of matrix metalloproteinase 2 (TIMP-2), and neutrophil gelatinase-associated lipocalin (NGAL) monomer are novel tubular kidney injury biomarkers. In multiple myeloma (MM), immunoglobulin free light chains (FLCs) play an integral role in renal impairment. This study aimed to investigate the correlation between new biomarkers and acclaimed parameters of renal failure, MM stage, and prognosis. *Materials and Methods*: The examined parameters included: urinary and serum cystatin-C, IGFBP-7, and TIMP-2, and urinary NGAL monomer in 124 enrolled patients. *Results*: Urinary and serum IGFBP-7 and urinary NGAL were higher among patients with an estimated glomerular filtration rate (eGFR) < 60 mL/min/1.73 m^2^, and positively correlated with urine light chains. Serum and urine IGFBP-7 and urine NGAL were greater among patients with a higher disease stage. In the whole study group, urinary concentrations of the studied markers were positively correlated with each other. In multiple linear regression, urinary IGFBP-7 and NGAL were associated with lower eGFR, independently of other urinary markers. *Conclusions*: Urinary IGFBP-7 and NGAL monomer may be useful markers of tubular renal damage in patients with MM. Biomarker-based diagnostics may contribute to earlier treatment that may improve renal outcomes and life expectancy in MM.

## 1. Introduction

Being one of the most common hematologic malignancies, particularly in the elderly patient, multiple myeloma (MM) represents a devastating disease that occurs due to uncontrolled proliferation of a plasma cell in bone marrow. Recent data suggests that with the aging of the general population, the prevalence of MM has increased [1]. Early mortality may not only be associated with disease progression, but also with concomitant organ involvement (i.e., renal failure) and therapy side-effects (i.e., infections) [2]. Production of monoclonal paraproteins, immunoglobulins (MIg) or free light chains (FLCs), is a characteristic feature of myeloma that is tied to its clinical manifestations. Although non-functional, paraproteins may be produced in excess, which alongside their physicochemical properties may precipitate the formation of lesions in vasculature and variably affect organ function. Owing to the differences between individual clones and associated MIg, kidney pathology is highly heterogeneous, with cast nephropathy as the dominant presentation [3]. Filtered excess FLCs undergo endocytosis in the tubular compartment, which may lead to a pattern of proximal injury, with promotion of proinflammatory and fibrotic pathways [4]. Up to half of incident MM cases may present with renal impairment, though the prevalence across studies depends on the chosen definition [5]. The importance of renal involvement, which may be reversible if treated early, is illustrated by data indicating a strong relationship with poor survival, particularly evident with the indication for renal replacement therapy (RRT) [6,7,8]. With the recent additions to the myeloma treatment armamentarium, newer drugs with increased efficacy against neoplastic plasmacytes have also been attributed with an improvement of the outcomes in kidney impairment [9,10]. However, real-world data indicate that there are cross-country inequities in the access to antimyeloma regimens and stem cell transplantation [11,12]. Therefore, earlier recognition of disease, including its end-organ manifestations, is necessary for timely initiation and optimization of therapy.

In practice, serum FLC and urinary albumin excretion may suggest the underlying nephropathological entity, though kidney biopsy remains the only tool that establishes a definitive diagnosis [13,14]. In some cases, renal manifestations in patients who do not fulfill diagnostic myeloma criteria may still necessitate antineoplastic treatment [14]. Recently, a multicenter study of myeloma kidney observed that nearly all cases (98%) presented with FLC levels > 500 mg/L at diagnosis, which has been advocated as the threshold by the International Kidney and Monoclonal Gammopathy working group [15,16]. FLCs are an important tool in diagnostic work-up. However, there is an unmet need for early and reliable diagnosis of kidney insult, ideally with a marker that represents the affected nephron compartment [17]. Marked changes in the routinely adopted measures of kidney function (i.e., creatinine, blood urea nitrogen) are reliant on a number of non-renal factors (comorbidity, muscle mass, medication, age) and are not an early and immediate indicator of renal injury. In the search for acute kidney injury molecules in the critically ill patient, several candidate urinary molecules released from injured tubular cells have emerged as very early and sensitive markers [18,19,20]. Researchers have also designed models that would account for the processes of renal injury occurring in myeloma [20].

Moreover, molecules that used to be associated only with AKI tubular injury markers are now under investigation as potential ones to predict the development of future chronic kidney injury. Identifying molecules that would aid in non-invasive diagnosis of particular patterns of MM-related renal lesions is a promising aspect of current efforts.

The aim of this study was to evaluate novel markers of kidney injury, insulin-like growth factor-binding protein 7 (IGFBP-7) and tissue inhibitor of matrix metalloproteinase 2 (TIMP-2), and examine their relationship with other emerging indicators of tubular damage (e.g., urinary neutrophil gelatinase-associated lipocalin, NGAL monomer, urinary cystatin C), as well as established measures of renal function (serum creatinine, cystatin C, and respective estimated glomerular filtration rate, eGFR) among MM patients with and without renal impairment (RI), and at various stages of MM progression. Serum concentrations of TIMP-2 and IGFBP-7 were assessed in part of the group to verify if the increased serum levels in MM may impact urinary concentrations of the markers.

## 2. Materials and Methods

### 2.1. Patients

This was a prospective observational study. Patients were recruited during ambulatory control visits at the Departments of Hematology of the University Hospital in Kraków, Poland between August 2016 and October 2017. Informed consent, age ≥18 years, and diagnosis of SMM or MM according to International Myeloma Working Group criteria were necessary for inclusion. A control group including 21 healthy volunteers (13 women, 8 men) with a median age of 50.6 years was recruited. Additionally, the concentrations of studied laboratory markers were assessed in 9 patients (7 women, 2 men; median age of 68.0 years) with MGUS to enable comparison with “overt MM” patients. Recent active infection; history of hepatitis B, C, or HIV; and neoplasms other than myeloma led to patient exclusion. Detailed history of disease was collected from all SMM/MM patients. At the initial study visit, data collection was focused on demographics, current and prior diagnoses, presence of bone lesions, and past and present treatment history (including response to treatment: complete response (CR), partial response (PR), stable disease (SD), progressive disease (PD)). The follow-up data were collected in October 2018 and included the date and cause of death, and the results of laboratory tests, including serum creatinine and eGFR, obtained at the final follow-up visit.

### 2.2. Ethics Statement

This study was conducted according to the principles of the Declaration of Helsinki and in compliance with the International Conference on Harmonization/Good Clinical Practice regulations. The study was approved by the Bioethics Committee of the Jagiellonian University and all patients signed an informed consent for their participation. All the patients were treated at the Department of Hematology, University Hospital in Krakow.

### 2.3. Laboratory Tests

On the day of blood collection, routine laboratory tests were performed and included complete blood counts, serum concentrations of creatinine, total protein, albumin, β2-microglobulin, lactate dehydrogenase (LDH), free light chains, and urine concentrations of light chains. Automatic hematological (Sysmex XE 2100 analyser, Sysmex, Kobe, Japan) and biochemical analyzers (Hitachi 917, Hitachi, Japan and Modular P, Roche Diagnostics, Mannheim, Germany) were used. The concentration of serum FLCs, urine light chains (LCs) κ and λ type, and β2-microglobulin were measured by the immunonephelometric method on a BN II analyser (Siemens GmbH, Munich, Germany). The determination of FLC κ and FLC λ was performed using Freelite reagents (Binding Site, Birmingham, UK). The immunophenotype of monoclonal protein was determined by serum immunofixation on agarose gel (EasyFix G26, Interlab, Rome, Italy).

Serum samples for other laboratory tests were aliquoted and stored in a temperature below −70 °C. These non-routine laboratory tests included urine NGAL, urine and serum cystatin-C, and IGFBP-7 and TIMP-2 in serum and urine. Interleukin-6 (IL-6), N-terminal B-type natriuretic propeptide (NT-proBNP), and serum concentrations of IGFBP-7 and TIMP-2 were measured in a subgroup of 73 patients.

The non-routine laboratory tests were performed in series, using commercially available immunoenzymatic (ELISA) test kits. Serum IL-6 was measured using Quantikine ELISA Human IL-6 Immunoassay (R&D Systems, Inc., Minneapolis, MN, USA). Serum and urine IGFBP-7 was measured using an IBP-7 ELISA Kit (EIAab Science Inc, Wuhan, China). Serum and urine TIMP-2 levels were measured using a Human Metalloproteinase inhibitor 2 ELISA Kit (EIAab Science Inc, Wuhan, China). Urine NGAL monomer was assessed using a Human NGAL monomer-specific ELISA Kit (BioPorto Diagnostics A/S, Hellerup, Denmark). Cystatin C concentrations in urine and serum were measured using Human Cystatin C ELISA (BioVendor Research and Diagnostic Products, Brno, Czech Republic). NT-pBNP concentrations in serum were measured by Enzyme-linked Immunosorbent Assay ELISA Kit For NT-ProBNP Human (Cloud-Clone Corporation, Huston, TX, USA).

The eGFR was calculated based on serum creatinine using the Chronic Kidney Disease—Epidemiology Collaboration (CKD-EPI)_Cr_ 2009 formula and based on serum cystatin C using the CKD-EPI_CysC_ 2012 formula [21].

### 2.4. Statistical Methods

The numbers of patients and percentages of the appropriate group were reported for categories and compared between groups using chi-squared test. Mean ± standard deviation was reported for normally distributed and median (IQR) for non-normally distributed quantitative variables. The Shapiro–Wilk’s test was used to assess normality. Between groups, the comparisons were done with the t-test, one-way analysis of variance (ANOVA), Mann-Whitney’s test, or Kruskal–Wallis test, depending on the number of groups and variables’ distributions. Pearson’s correlation coefficient was calculated for simple correlations, and linear multiple regression was used to assess the independent predictors of selected variables, as described in the text of the results. Right-skewed variables were log-transformed before the analyses. Multiple logistic regression was used to study the association between urinary biomarkers and eGFR values < 60 mL/min/1.73 m^2^. Simple Cox proportional hazard regression was used to assess the relationships between studied laboratory markers and overall survival, calculated from the date of the initial study visit and the date of death or last follow-up assessment, whatever occurred first. Simple logistic regression was used to evaluate the association between the studied markers and the presence of renal impairment at the end of follow-up (final eGFR < 60 mL/min/1.73 m^2^). The statistical tests were two-tailed and *p* < 0.05 indicated statistical significance. Statistica 12.0 (StatSoft, Tulsa, OK, USA) was used for computations.

## 3. Results

### 3.1. Patient Characteristics

The study enrolled 124 MM patients, 73 women and 51 men, with a mean ± standard deviation age of 66 ± 10 years and median (interquartile range, IQR) time from MM diagnosis of 30 (14; 63) months. There were 7 (6%) patients with smoldering MM (SMM), 80 (65%) with International Staging System (ISS) stage 1, 22 (18%) with stage 2, and 15 (12%) with stage 3 MM. Patients were at different treatment stages: 115 patients (93%) underwent at least one line of chemotherapy treatment; of those, 58 patients were receiving maintenance undergoing chemotherapy at the time of the study visit. Fifty-eight patients (47%) underwent autologous peripheral blood stem cell transplantation (auto-PBSCT) as a part of MM treatment. In the studied group, 9 (7%) patients had a history of acute kidney injury (AKI).

Among 124 patients, the mean ± standard deviation eGFR was 71 ± 25 mL/min/1.73 m^2^ and 28 (23%) patients had eGFR ≤60 mL/min/1.73 m^2^. Patients with eGFR values below 60 mL/min/1.73 m^2^ were older, had more advanced disease (higher ISS stage), and achieved complete response (CR) less often when compared to patients with eGFR > 60 mL/min/1.73 m^2^. History of AKI was more common in these patients (Table 1). Higher concentrations of serum β2-microglobulin, serum free light chains, and urine light chains λ; lower serum albumin and blood hemoglobin; more frequent proteinuria; and higher serum interleukin-6 and NT-proBNP levels were associated with eGFR ≤ 60 mL/min/1.73 m^2^ (Table 2). History of AKI was more common in these patients.

### 3.2. Associations between Urinary Markers of Tubular Injury and Clinical Characteristics of MM Patients

In the simple comparison, the concentrations of studied urinary markers of tubular injury did not differ between patients with SMM and symptomatic MM. Neither the history of chemotherapy (untreated versus treated patients, number of prior treatment schemes) nor the effects of treatment (CR, PR, SD, or PD) were associated with the urinary concentrations of the studied markers.

Patients with symptomatic MM who were receiving maintenance chemotherapy at the time of sample collection had higher urinary IGFBP-7 concentrations (median 7.73 ng/mL; IQR 2.88; 23.38 ng/mL) compared to the rest of the group (median 3.81 ng/mL; IQR 2.08; 9.25 ng/mL; *p* = 0.014). Moreover, lower urinary IGFBP-7 levels were observed in patients who underwent autologous peripheral blood stem cell transplantation (auto-PBSCT) (median 4.67 ng/mL; IQR 2.08; 8.33 ng/mL) as compared to those who did not (median 8.51 ng/mL; IQR 3.18; 21.00 ng/mL; *p* = 0.009). No association was observed between the studied tubular markers and patients’ age or the time from the diagnosis of MM.

To examine the association between the studied urinary markers and advancement of monoclonal gammopathy, we show the results obtained from pre-malignant monoclonal gammopathy of undetermined significance (MGUS), SMM, and MM subjects (Figure 1 and Figure 2). As illustrated by the figures, neither of the studied urinary markers were significantly correlated with the spectrum of monoclonal gammopathy, ranging from MGUS to advanced MM. In patients with symptomatic MM (*n* = 117), urine IGFBP-7 and urine NGAL monomer positively correlated with ISS stage (R = 0.46; *p* < 0.001 and R = 0.34; *p* < 0.001, respectively); however, this was in line with decreasing eGFR values (Figure 1A,B).

For comparison, the markers of tubular injury were measured in the urine of 21 healthy individuals. Only the concentrations of urinary IGFBP-7 were significantly higher in the studied MM patients than in healthy controls (median 2.59; lower; upper quartile 2.60; 4.09 ng/mL; *p* = 0.011). As shown in Figure 2B, the urinary concentrations of IGFBP-7 in the majority of MM patients in ISS stages II and III exceeded the upper concentrations observed in healthy individuals.

### 3.3. Correlations of Urinary Markers of Tubular Injury among Patients with MM

As shown in Table 3, urinary concentrations of IGFBP-7 and NGAL in the studied patients with MM were significantly positively correlated with serum creatinine and cystatin C, and consequently, negatively correlated with the respective eGFR values. Positive correlations were observed between IGFBP-7 and NGAL and the concentrations of immunoglobulin light chains in urine. Moreover, these two markers positively correlated with serum β2-microglobulin and negatively with serum albumin. To the contrary, urinary TIMP-2 and cystatin C did not correlate significantly with the above-mentioned variables (Table 3).

The studied urinary markers of tubular injury did not correlate with serum free light chains, apart from urine NGAL monomer, and none of them correlated with urine light chains. Additionally, there were no significant correlations between the markers and serum interleukin 6. Urinary IGFBP-7 and NGAL were negatively correlated with blood hemoglobin (R = −0.25; *p* = 0.005; and R = −0.20; *p* = 0.023, respectively). Moreover, urinary IGFBP-7 positively correlated with serum NT-proBNP (R = 0.26; *p* = 0.026).

In the studied patients with MM, urinary markers of tubular injury were interrelated (we report Pearson correlation coefficients after log-transformation of the variables): urinary cystatin C correlated positively with NGAL monomer (R = 0.24; *p* = 0.008), IGFBP-7 (R = 0.29; *p* = 0.001), and TIMP-2 (R = 0.27; *p* = 0.002); NGAL correlated positively with IGFBP-7 (R = 0.41; *p* < 0.001) and TIMP-2 (R = 0.20; *p* = 0.024); and IGFBP-7 correlated positively with TIMP-2 (R = 0.29; *p* = 0.001). No association was observed between TIMP-2 concentrations in the serum and urine (*p* > 0.9). To the contrary, IGFBP-7 concentrations in the serum and urine were significantly correlated (R = 0.24; *p* = 0.041).

For the product of urinary concentrations of IGFBP-7 and TIMP-2, we observed the same (although weaker) set of associations as for urinary IGFBP-7 alone.

In the multiple linear regression, the negative correlations between urinary NGAL and IGFBP-7 and eGFR were independent of each other and of the other studied urinary markers of tubular injury (Table 4A). However, after adjustment for confounders (age, smoldering versus symptomatic MM, ISS, LDH, remission versus stable or progressive disease, and the urinary concentrations of involved light chains), only urinary NGAL was independently associated with eGFR (Table 4B). Additionally, in the logistic regression, urinary NGAL significantly predicted renal insufficiency (eGFR < 60 mL/min/1.73 m^2^) independently of clinically relevant patient characteristics (odds ratio per 1 ng/mL increase in the NGAL concentration was 1.01; 95% confidence interval 1.001–1.03; *p* = 0.025).

### 3.4. Associations between Studied Markers of Tubular Injury and Follow-Up Data

We collected the follow-up data on mortality and renal function up to 27 months from the start of the study. The median observation time was 21 months (IQR 16–24 months). The data on mortality were available for 123 patients. During the follow-up, 23 patients (19%) died. The major causes of death were MM (12 patients) and infection (7 patients). We did not observe statistically significant associations between the studied markers (urinary concentrations of NGAL monomer, cystatin C, TIMP-2 and IGFBP-7; serum TIMP-2 and IGFBP-7) and all-cause mortality (*p* > 0.05 in simple Cox proportional hazard regression).

Data on eGFR at the end of follow-up were available for 118 patients (95% of the studied group). Median (IQR) eGFR at the end of follow-up was 73 (51; 87) mL/min/1.73 m^2^. Significant correlations were observed between final eGFR values and log-transformed concentrations of urine NGAL monomer (R = −0.34; *p* < 0.001), urine IGFBP-7 (R = −0.37; *p* < 0.001), serum TIMP-2 (R = −0.29; *p* = 0.012), and serum IGFBP-7 (R = −0.42; *p* < 0.001). Among 91 patients with initial eGFR above 60 mL/min/1.73 m^2^, there were 20 who developed renal impairment during the follow-up defined as final eGFR <60 mL/min/1.73 m^2^. Higher urinary concentrations of IGFBP-7 significantly predicted the renal impairment in those patients (odds ratio per 1 ng/mL increase in urine IGFBP-7 equaled 1.04; 95% confidence interval 1.001–1.08; *p* = 0.039).

## 4. Discussion

To the best of our knowledge, this is the first study evaluating the relationship of novel markers of tubular injury with measures of RI in healthy volunteers and a sample of MM patients at different stages of disease and treatment. The salient finding of this study is the association of urinary IGFBP-7 with renal impairment and tumor burden (marked by FLCs, albumin, β2-microglobulin). Urinary IGFBP-7 concentrations were significantly elevated in symptomatic patients undergoing maintenance chemotherapy and correlated with the ISS. Our MM sample may reflect the population encountered in routine practice and thus inform on the utility of novel RI biomarkers in the clinic. With regard to future prospects, this study supports further investigation of IGFBP-7 in the development of a conceptual model of renal injury in MM.

Early studies outlined a relationship between serum proteins from the IGFBP family and decreased glomerular filtration in the adult population [19]. Urine IGFBP-7 and TIMP-2 have also been identified as highly sensitive markers of kidney injury in critically ill patients, exerting G_1_ cell-cycle arrest in renal tubular cells, which may be an inherent mechanism of early response to kidney insult [18]. On the other hand, studies in murine models have favored a mechanism of increased permeability, filtration, and impaired reabsorption, rather than stress-induced gene transcription [22]. Studies have focused on the assay of either urinary IGFBP-7 or the composite of urinary IGFBP-7 and TIMP-2 in the setting of AKI [23,24,25]. These novel molecules appear to be favorable biomarkers in predicting adverse outcomes, renal recovery, or the development and progression of CKD [26]. The utility of the IGFBP-7 and TIMP-2 assay in multiple myeloma is unknown, and our findings serve as a preliminary report on their relationship with renal impairment.

Interestingly, high IGFBP-7 expression in MM cells has been considered a marker of high-risk disease due to its association with poor survival and adverse chromosomal aberrations [27]. We observed that both serum and urinary IGFBP-7 are markedly higher in patients with RI. However, the significance and source of this molecule in circulation remains unclear. Moreover, IGFBP-7 levels were increased in patients remaining on maintenance chemotherapy and were reduced in patients with a history of stem cell transplant. IGFBP-7 may antagonize the inhibitory effects on myeloma survival promoted by bone morphogenetic proteins [27,28], which may provide a rationale for its relationship with tumor load and symptomatic disease.

IGFBP-7 appears to be secreted, mainly in the proximal tubule compartment, and may reflect an autocrine or paracrine response to prevent cellular proliferation following tissue injury [29]. It should be noted that the available data on the origin and source in renal compartment are preliminary and IGFBP-7 remains only a candidate molecule in the setting of nephropathy outside of acute care. Expression in the renal tissue may hold a relationship with primary disease (e.g., diabetic nephropathy) and is under investigation [30,31]. Although certain biomarkers can be conveniently extrapolated based on their utility in the general population, validation in diseases with a distinct mechanism of kidney injury is necessary [32].

NGAL is produced and released during granulocyte maturation in bone marrow by polymorphonuclear myeloid-derived suppressor cells (PMN-MDSCs). The dimer form is the major molecular form of free NGAL secreted by neutrophils and its overexpression is observed in blood cells from patients with all types of leukemia [33,34] We observed that urinary NGAL is an independent predictor of glomerular filtration (based on CKD-EPI formulas for eGFR) and holds associations with several established prognostic markers of tumor burden (FLCs, albumin, β2-microglobulin). In an effort to reduce the interreference of neutrophil production, our assay was based on the monomeric form of NGAL (that is specifically associated with tubular epithelial stress), rather than the dimeric molecule, with the renal tubular epithelium as the purported site of origin. After adjustment for age and disease characteristics, urinary NGAL, rather than IGFPB-7, was independently associated with eGFR and appears to be a robust predictor of RI. A recent meta-analysis underscores the relationship between NGAL and glomerular filtration in chronic kidney disease, particularly in more advanced stages [33]. One series on MM found serum NGAL and cystatin C to be very sensitive indicators of RI in incident cases of MM, which may be reflective of abnormal tubular and glomerular function. However, urinary molecules were not evaluated at the time [35]. According to some authors, urinary NGAL may be a more favorable marker of RI in MM than the serum concentrations. The rationale has been drawn from higher urinary NGAL concentrations in patients with renal involvement (as opposed to non-renal MM), correlations with eGFR and urinary FLC, and greater sensitivity for renal injury over serum NGAL [36]. NGAL levels may be elevated early in the asymptomatic stages of the spectrum of monoclonal gammopathy (as opposed to, e.g., cystatin C, which rises in symptomatic MM) [35]. Noteworthy, NGAL in different hematologic malignancies was associated with vascular endothelial growth factor (VEGF), and soluble receptor for advanced glycation end-products (sRAGE) in bone marrow [37]. Moreover, angiogenic switching is a key process during transition from premalignant asymptomatic MGUS to developed MM and is considered to predict MM progression. One of the stimulating MM-associated angiogenesis proteins is the junctional adhesion molecule-A (JAM-A), a potential target for future MM therapy [38]. Similarly, the IGFBPs family in the bone marrow microenvironment plays an important role in progression and treatment resistance in MM, though the extracellular IGF system in MGUS and MM did not include IGFBP-7 [39].

Interestingly, mechanisms involved in IGFBP-7 upregulation seem to be different than for NGAL, which may suggest that these two molecules are reflective of different pathways of response to kidney injury [29]. In our study, these two markers have been associated with eGFR independently, supporting the latter theory. This finding also suggests that combining biomarkers may be superior to a singular measure.

TIMP-2 is a novel molecular biomarker involved in tumorigenesis and development of MM by suppressing tumor cell proliferation and metastasis [40]. However, its increase was also observed in various nephropathies including diabetic, glomerulosclerosis, tubulointerstitial fibrosis, and vasculitis [40,41]. The distal tubule is a major source of TIMP-2 [29], though extra-renal sources related to the progression of myeloma are also possible.

Our data indicate that tubular injury biomarkers are interrelated, namely urinary IGFBP-7 and TIMP-2 with urinary cystatin C and urinary NGAL. We also observed significant correlations between IGFBP-7 concentrations in serum and urine but not for TIMP-2. We also noted similar patterns for the product of urinary IGFBP-7/TIMP-2, as well as the singular urinary IGFBP-7 assay. So far, the available data for TIMP-2/IGFBP-7 suggest its utility as a sensitive AKI biomarker [18,22,23,24,25]. In MM, the composite marker does not hold favorable value, which may stem from the involvement of TIMP-2 in pathogenesis, or the chronic character of injury in MM. The reported intercorrelations between tubular kidney injury biomarkers may indicate a future possibility of biomarker models to reflect the etiology and nephropathological diagnosis. Considering the three main types of paraprotein-related renal lesions in MM (MCN, amyloid light-chain (AL amyloidosis), and monoclonal immunoglobulin deposition disease (MIDD)), future studies assessing the relationship of biomarkers with biopsy evidence are of interest. Although we did not observe urinary TIMP-2 to correlate with eGFR or FLCs, other studies have focused on TIMPs in MIDD and AL amyloidosis, revealing different roles in pathogenesis, even though both diseases are caused by FLC deposition [42,43]. When investigating TIMP-2 as a kidney injury biomarker in myeloma, it is necessary to consider the involvement in tumorigenesis, disease progression, and formation of osteolytic lesions [44,45]. Indeed, increased production of TIMP-2 in bone marrow stromal cells may play a pivotal role in the development of osteolytic lesions in MM [46,47]. This may reflect weaker correlations of TIMP-2 with eGFR.

Due to the numerous processes involved in the pathogenesis and progression of MM and its kidney involvement, it is likely that a singular biomarker will be insufficient for an adequate understanding of ongoing tissue damage. Biomarker models based on several molecules accounting for the complexity of disease may prove to hold favorable diagnostic and prognostic value [48].

There are apparent limitations in the retrospective character of this single-center study, which involves a heterogenous population, and investigated established rather than incident cases of MM, which were subject to different treatment regimens. Moreover, in this study, we used the Multiple Myeloma International Staging System (ISS) instead of the Revised Multiple Myeloma International Staging System (R-ISS) due to limited results of chromosomal abnormalities performed by iFISH (15 patients out of the group of 124). The accessibility to proper treatment is limited; therefore, some patients are administered drug programs, which are not in line with mSMART tool (Mayo Stratification for Myeloma and Risk-Adapted Therapy) and do not include the need to define the genetic and biologic features of MM.

However, this study provides a wide array of data on the utility of novel tubular injury measures, and their relationship with clinically meaningful patient characteristics, which may allow for their future study and incorporation in models reflective of the processes ongoing in the myeloma kidney.

## 5. Conclusions

Urinary IGFBP-7 and NGAL monomer may be useful markers of tubular renal damage in patients with MM and may be considered as predicative of future chronic kidney development. Biomarker-based diagnostics may contribute to earlier treatment that may improve renal outcomes and life expectancy in MM.

## Figures and Tables

**Figure 1 medicina-57-01348-f001:**
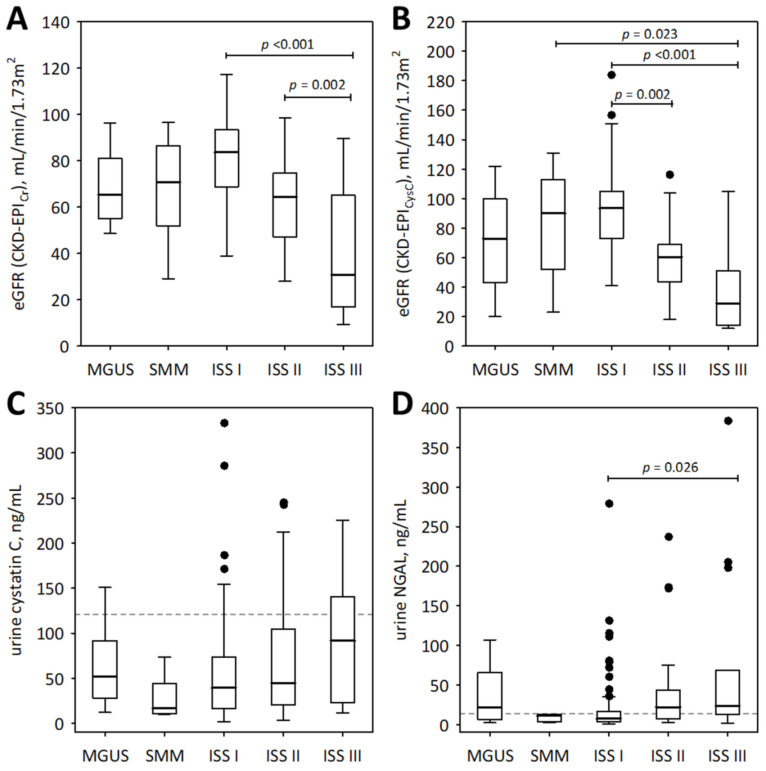
The values of eGFR based on serum creatinine (**A**) and cystatin C (**B**), and urinary concentrations of cystatin C (**C**) and NGAL monomer (**D**) among patients with monoclonal gammopathy ranging from MGUS to symptomatic MM. Data are shown as the median (central line), interquartile range (box), and non-outlier range (whiskers). The outliers are shown as points. We report *p*-values for significant differences between groups (calculated in post-hoc tests following the Friedman ANOVA). The dotted lines in panels (**C**,**D**) indicate non-outlier maximum urinary concentrations of the markers obtained in the control group of 21 healthy individuals.

**Figure 2 medicina-57-01348-f002:**
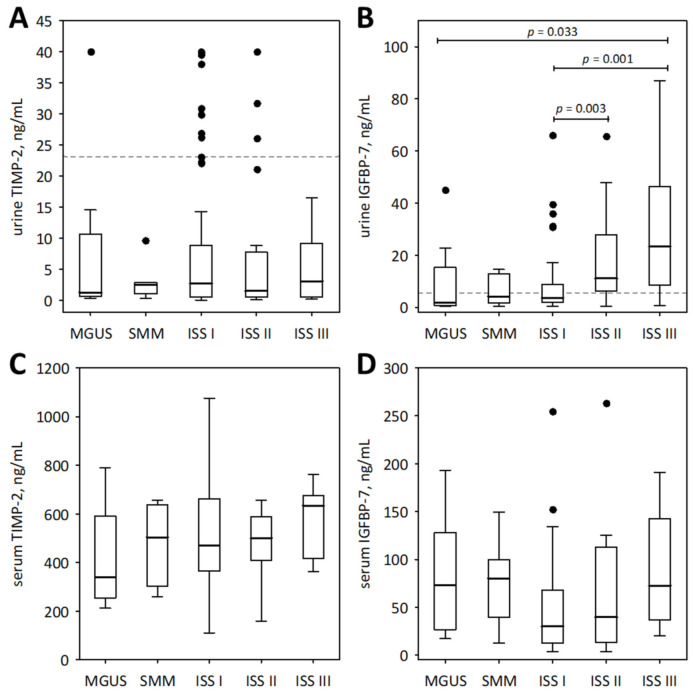
Urinary (**A**,**B**) and serum (**C**,**D**) concentrations of TIMP-2 (**A**,**C**) and IGFBP-7 (**B**,**D**) among patients with monoclonal gammopathy ranging from MGUS to symptomatic MM. Serum concentrations of the markers were assessed in a subgroup of 73 patients. Data are shown as the median (central line), interquartile range (box), and non-outlier range (whiskers). The outliers are shown as points. We report *p*-values for significant differences between groups (calculated in post-hoc tests following the Friedman ANOVA). The dotted lines in panels (**A**,**B**) indicate non-outlier maximum urinary concentrations of the markers obtained in the control group of 21 healthy individuals.

**Table 1 medicina-57-01348-t001:** Clinical characteristics of the studied patients with MM.

Variable	eGFR (CKD-EPI_Cr_) ≤ 60 mL/min/1.73 m^2^ (*n* = 28)	eGFR (CKD-EPI_Cr_) > 60 mL/min/1.73 m^2^ (*n* = 96)	*p*
Mean age ± standard deviation, years	72 ± 9	65 ± 10	0.001
Male sex, *n* (%)	10 (36)	41 (43)	0.5
Median time since diagnosis (IQR), months	51 (13; 86)	28 (14; 57)	0.2
Smoldering MM	2 (7)	5 (5)	0.7
ISS			<0.001
Stage I, *n* (%)	8 (29)	79 (82)	
Stage II, *n* (%)	10 (36)	12 (12)	
Stage III, *n* (%)	10 (36)	5 (5)	
Immunofixation			
IgG, *n* (%)	22 (79)	67 (70)	0.4
IgM, *n* (%)	1 (4)	1 (1)	0.3
IgA, *n* (%)	5 (18)	20 (21)	0.7
κ, *n* (%)	17 (61)	60 (62)	0.9
λ, *n* (%)	12 (43)	33 (34)	0.4
Free light chains, *n* (%)	6 (21)	12 (12)	0.2
Biclonal, *n* (%)	2 (7)	3 (3)	0.3
Non-secretory MM, *n* (%)	0	4 (4)	0.6
Disease state			0.036
CR, *n* (%)	4 (14)	40 (42)	
PR, *n* (%)	15 (54)	29 (30)	
SD, *n* (%)	3 (11)	6 (6)	
PD, *n* (%)	6 (21)	21 (22)	
Number of prior treatment schemes			0.3
No treatment, *n* (%)	2 (7)	7 (7)	
1, *n* (%)	5 (18)	29 (30)	
2, *n* (%)	10 (36)	31 (32)	
3 and more, *n* (%)	11 (39)	29 (30)	
On chemotherapy treatment at the time of samples collection, *n* (%)	16 (57)	42 (44)	0.2
History of auto-PBSCT, *n* (%)	7 (25)	51 (53)	0.009
Bone lesions, *n* (%)	17 (61)	58 (60)	0.9
History of AKI, *n* (%)	6 (21)	3 (3)	0.001

Abbreviations: AKI, acute kidney injury; autoPBSCT, autologous Peripheral Blood Stem Cell Transplantation; CKD-EPI_Cr_, Chronic Kidney Disease Epidemiology Collaboration equation based on serum creatinine; CR, complete response; eGFR, estimated glomerular filtration rate; Ig, immunoglobulin; ISS, International Staging System; IQR, interquartile range; MM, multiple myeloma; PD, progressive disease; PR, partial response; SD, stable disease.

**Table 2 medicina-57-01348-t002:** The results of the laboratory tests of the studied patients with MM. Data are shown as median (IQR) or mean ± standard deviation.

	eGFR (CKD-EPI_Cr_) ≤ 60 mL/min/1.73 m^2^ (*n* = 28)	eGFR (CKD-EPI_Cr_) > 60 mL/min/1.73 m^2^ (*n* = 96)	*p*
Serum creatinine, µmol/L	124 (104; 218)	73 (65; 81)	<0.001
eGFR (CKD-EPI_Cr_), mL/min/1.73 m^2^	43 (23; 50)	84 (71; 93)	NA
Serum cystatin C, mg/L	1.55 (1.10; 2.73)	0.83 (0.70; 1.01)	<0.001
eGFR (CKD-EPI_CysC_), mL/min/1.73 m^2^	39 (18; 60)	93 (72; 105)	<0.001
Serum albumin, g/L	40.0 (37.0; 42.1)	44.0 (40.6; 45.6)	<0.001
Serum β2-microglobulin, mg/L	4.51 (3.23; 7.34)	2.40 (2.01; 3.06)	<0.001
Serum FLC κ, mg/L	38.0 (15.0; 91.8)	17.6 (11.6; 37.7)	0.010
Serum FLC λ, mg/L	24.0 (14.0; 52.9)	16.0 (11.4; 22.4)	0.023
Involved serum FLC	53.4 (30.5; 116.0)	22.6 (15.4; 91.6)	0.022
Urine LC κ, mg/L	11.8 (ND; 49.9)	ND (ND; 26.8)	0.2
Urine LC λ, mg/L	6.1 (ND; 15.0)	ND (ND; 4.4)	0.001
Involved urine LC	11.8 (6.8; 43.6)	6.8 (5.01; 27.8)	0.06
Leukocyte count, ×10^3^/µL	6.36 (5.44; 7.40)	5.66 (4.53; 7.06)	0.1
Hemoglobin, g/dL	11.7 ± 1.7	12.9 ± 1.6	0.006
Platelet count, ×10^3^/µL	176 (149; 247)	171 (141; 208)	0.4
Lactate dehydrogenase, U/L	388 (320; 418)	351 (303; 404)	0.1
Serum interleukin 6, pg/mL ^a^	5.26 (1.76; 7.45)	2.43 (1.56; 4.42)	0.039
Serum NT-proBNP, pg/mL ^a^	178 (44; 460)	56 (30; 201)	0.008
Proteinuria, *n* (%)	13 ± 46	13 ± 14	<0.001
Urine IGFBP-7, ng/mL	11.83 (5.57; 31.84)	4.81 (2.15; 9.55)	<0.001
Urine TIMP-2, ng/mL	2.32 (0.57; 8.77)	2.55 (0.47; 8.10)	0.9
Urine TIMP-2 × IGFBP-7, ng^2^/mL^2^	26.20 (3.31; 205.75)	9.02 (1.01; 69.4)	0.08
Urine cystatin C, ng/mL	55.9 (19.8; 131.4)	35.4 (16.1; 76.3)	0.1
Urine NGAL monomer, ng/mL	25.2 (8.1; 71.6)	8.2 (4.0; 17.1)	<0.001
Serum IGFBP-7, ng/mL ^a^	97.5 (41.2; 128.0)	33.0 (11.6; 61.5)	<0.001
Serum TIMP-2, ng/mL ^a^	628 (462; 663)	445 (363; 614)	0.029
Urine/serum IGFBP-7 ^a^	0.117 (0.050; 0.440)	0.160 (0.083; 0.463)	0.6
Urine/serum TIMP-2 ^a^	0.0047 (0.0012; −0.0237)	0.0036 (0.0009; 0.0160)	0.6

^a^ Results available in 73 patients, including 23 with eGFR ≤60 mL/min/1.73 m^2^ and 50 with eGFR >60 mL/min/1.73 m^2^. Abbreviations: CKD-EPICr, Chronic Kidney Disease Epidemiology Collaboration equation based on serum creatinine; CKD-EPICysC, Chronic Kidney Disease Epidemiology Collaboration equation based on serum cystatin C; eGFR, estimated glomerular filtration rate; FLC, free light chains; IGFBP-7, insulin-like growth factor-binding protein 7; IQR, interquartile range; LC, light chains; NA, not applicable; ND, not detectable; NGAL, neutrophil gelatinase-associated lipocalin; NT-proBNP, N-terminal pro-brain natriuretic peptide; TIMP-2, tissue inhibitor of matrix metalloproteinase 2.

**Table 3 medicina-57-01348-t003:** Correlations between the studied urinary markers of tubular injury and the measures of glomerular filtration, serum albumin, β2-microglobulin, and urinary concentrations of light chains among 124 studied patients with MM. Pearson correlation coefficients were calculated after log-transformation of right-skewed variables.

	log (Urine IGFBP-7)		log (Urine TIMP-2)		log (Urine Cystatin C)		log (Urine NGAL Monomer)	
	R	*p*	R	*p*	R	*p*	R	*p*
log (serum creatinine)	0.34	<0.001	−0.10	0.3	0.20	0.027	0.38	<0.001
eGFR (CKD-EPI_Cr_)	−0.38	<0.001	−0.02	0.8	−0.14	0.1	−0.34	<0.001
log (serum cystatin C)	0.38	<0.001	−0.07	0.5	0.07	0.5	0.28	0.002
eGFR (CKD-EPI_CysC_)	−0.40	<0.001	0.05	0.6	−0.03	0.7	−0.28	0.002
Serum albumin	−0.32	<0.001	−0.11	0.2	−0.17	0.054	−0.28	0.001
log (serum β2-microglobulin)	0.48	<0.001	0.01	0.9	0.14	0.1	0.38	<0.001
log (urine light chains κ)	0.30	0.001	0.14	0.1	0.13	0.1	0.26	0.003
log (urine light chains λ)	0.28	0.002	−0.04	0.6	0.15	0.1	0.33	<0.001
log (involved urine light chains)	0.23	0.009	0.12	0.2	0.12	0.2	0.20	0.028

Abbreviations: CKD-EPICr, Chronic Kidney Disease Epidemiology Collaboration equation based on serum creatinine; CKD-EPICysC, Chronic Kidney Disease Epidemiology Collaboration equation based on serum cystatin C; eGFR, estimated glomerular filtration rate; IGFBP-7, insulin-like growth factor-binding protein 7; NGAL, neutrophil gelatinase-associated lipocalin; TIMP-2, tissue inhibitor of matrix metalloproteinase 2.

**Table 4 medicina-57-01348-t004:** Multiple linear model showing which of the studied urinary markers of tubular injury predicts eGFR in the studied group of MM patients (*n* = 124): A. independently of each other; B. independently of clinically relevant covariates.

Independent Variable	Standardized Beta ± Standard Error	*p*
A		
log (urine IGFBP-7)	−0.31 ± 0.09	0.001
log (urine TIMP-2)	0.12 ± 0.09	0.2
log (urine cystatin C)	−0.03 ± 0.09	0.8
log (urine NGAL monomer)	−0.23 ± 0.09	0.012
R^2^ for the model	0.20	<0.001
B		
log (urine IGFBP-7)	−0.02 ± 0.07	0.7
log (urine NGAL monomer)	−0.14 ± 0.07	0.046
Age	−0.38 ± 0.07	<0.001
Symptomatic MM	0.03 ± 0.06	0.6
ISS stage II	−0.29 ± 0.07	<0.001
ISS stage III	−0.48 ± 0.07	<0.001
LDH above upper reference limit	−0.09 ± 0.06	0.2
No remission	0.08 ± 0.07	0.2
log (involved urinary light chains)	0.06 ± 0.07	0.4
R^2^ for the model	0.57	<0.001

Abbreviations: IGFBP-7, insulin-like growth factor-binding protein 7; ISS, International Staging System; LDH, lactate dehydrogenase; MM, multiple myeloma; NGAL, neutrophil gelatinase-associated lipocalin; TIMP-2, tissue inhibitor of matrix metalloproteinase 2.

## Data Availability

The data is available from the corresponding author upon reasonable request.

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
