# Peer review of "Renal Impairment Detectors: IGFBP-7 and NGAL as Tubular Injury Markers in Multiple Myeloma Patients"

_medicina, 2021, doi:10.3390/medicina57121348_

Round 1

Reviewer 1 Report

medicina-1486103

Early detectors of renal impairment: IGFBP-7 and NGAL as tubular injury markers in multiple myeloma patients

The article " Early detectors of renal impairment: IGFBP-7 and NGAL as tubular injury markers in multiple myeloma patients" (medicina-1486103) by Woziwodzka K, et al. demonstrated that IGFBP-7 and NAGL were associated with renal insufficiency in MM patients. I considered that IGFBP-7 and NGAL were new biomarkers for prognosis and might detect AKI in MM patients. However, the author did not demonstrate that IGFBP-7 and NGAL could predict AKI early although this article of title was “Early detectors of renal impairment: IGFBP-7 and NGAL as tubular injury markers”. Thus, there were several issues to be addressed as below.

  1. The authors did not demonstrate that IGFBP-7 and NGAL could detect AKI early although IGFBP-7 and NGAL was significantly associated with current renal damage. If you showed that, relationship between the IGFBP-7 and NGAL at base line and change of renal function should be analyzed.

  1. According to a previous article, IGFBP-7 predicted short survival time in MM patients. Were IGFBP-7 and NGAL associated with survival time in this study?

  1. The patients who were on treatment was included in this study. Did treatment affect IGFBP-7 and NGAL level?

  1. The author mentioned that it was interesting issue that IGFBP-7 and NGAL divided into MM related AKI and non-MM related AKI. I agree with your opinion. Could you try to demonstrate that IGFBP-7 and NGAL was associated with MM related renal damage, such as MCN, or non-MM related renal damage in this study cohort? It was stringently difficult to categorized into MM and non-MM related renal damages, meanwhile I considered that renal damage with BJP might be identified as MM related renal damage is this situation.

Author Response

Response to Reviewer 1 

The authors thank the Reviewer for a thorough evaluation of the manuscript. We have carefully addressed the comments of both Reviewers. Below, we present the detailed answers to the Reviewer’s comments and the description of the modifications introduced upon revision of the manuscript.

The article "Early detectors of renal impairment: IGFBP-7 and NGAL as tubular injury markers in multiple myeloma patients" (medicina-1486103) by Woziwodzka K, et al. demonstrated that IGFBP-7 and NGAL were associated with renal insufficiency in MM patients. I considered that IGFBP-7 and NGAL were new biomarkers for prognosis and might detect AKI in MM patients. However, the author did not demonstrate that IGFBP-7 and NGAL could predict AKI early although this article title was “Early detectors of renal impairment: IGFBP-7 and NGAL as tubular injury markers”. Thus, there were several issues to be addressed as below.

Response: Recent data about tubular injury markers focuses on their relation with AKI. However, tubular epithelial injury contributes to chronic kidney disease in future and this main issue was investigated in our study. Only several research and review articles emphasize this new perspective on this matter. Our aim was to evaluate the associations between “so far used to considered as only AKI markers” molecules like NGAL and IGFBP-7 with chronic process of renal impairment and their ability to predict chronic loss of eGFR. We have emphasized this also in introduction by adding sentence (lines 85-86).

To avoid misunderstanding the title with this relationship to AKI, the change to  „Renal impairment detectors:  IGFBP-7 and NGAL as tubular injury markers in multiple myeloma patients”.

  1. The authors did not demonstrate that IGFBP-7 and NGAL could detect AKI early although IGFBP-7 and NGAL was significantly associated with current renal damage. If you showed that, relationship between the IGFBP-7 and NGAL at base line and change of renal function should be analyzed.

Response: Our main aim was to find novel perspective on these markers, as ones that can predict the risk of chronic kidney disease development, not AKI.

As a part of our study, we collected the follow-up data, including the data on mortality and serum creatinine / eGFR values at the end of observation. The follow-up data regarding mortality were available for 123 patients, while the data on eGFR at the end of the follow-up were available for 118 patients (95%). The median observation time was 21 months (IQR 16-24 months). Significant correlations were observed between log-transformed concentrations of urine NGAL monomer (R=-0.34; P<0.001), urine IGFBP-7 (R=-0.37; P<0.001), serum TIMP-2 (R=-0.29; P=0.012) and serum IGFBP-7 (R=-0.42; P<0.001) and final eGFR values. Among 91 patients with initial eGFR above 60 ml/min/1.73 m2, there were 20 who developed renal impairment during the follow-up defined as final eGFR <60 ml/min/1.73 m2. Higher urinary concentrations of IGFBP-7 significantly predicted the renal impairment in those patients (odds ratio per 1 ng/ml increase in urine IGFBP-7 equaled 1.04; 95% confidence interval 1.001-1.08; P=0.039). We have added this information in paragraph 3.4, lines 285-293.

  1. According to a previous article, IGFBP-7 predicted short survival time in MM patients. Were IGFBP-7 and NGAL associated with survival time in this study?

Response: We have not observed any significant associations between the studied markers (including serum and urine concentrations of IGFBP-7) and mortality. This information has been added in paragraph 3.4, lines 278-284.

  1. The patients who were on treatment was included in this study. Did treatment affect IGFBP-7 and NGAL level?

Response: We did not observe the association between urinary concentrations of NGAL monomer and the ongoing maintenance treatment (). On the contrary, patients with symptomatic MM who were receiving maintenance chemotherapy at the time of samples collection had higher urinary IGFBP-7 concentrations (median 7.73 ng/ml; IQR 2.88; 23.38 ng/ml) compared to the rest of the group (median 3.81 ng/ml; IQR 2.08; 9.25 ng/ml; P = 0.014). This latter information is included in the manuscript text, paragraph 3.2, lines 194-197. Other studied markers were not associated with ongoing maintenance treatment.

  1. The author mentioned that it was interesting issue that IGFBP-7 and NGAL divided into MM related AKI and non-MM related AKI. I agree with your opinion. Could you try to demonstrate that IGFBP-7 and NGAL was associated with MM related renal damage, such as MCN, or non-MM related renal damage in this study cohort? It was stringently difficult to categorized into MM and non-MM related renal damages, meanwhile I considered that renal damage with BJP might be identified as MM related renal damage is this situation.

Response: The best undisputed option to categorize patients into MM-related AKI and non-MM related AKI is kidney biopsy, similarly to the same diversification in case of chronic changes. The idea of  investigation the panels of several biomarkers indicating the MM-related changes and damaged parts of nephrons seems to be future focus. In our study we checked the tubular injury markers associations with FLC (Table 3) and comment them (“The studied urinary markers of tubular injury did not correlate with serum free light chains”) and extend the explanation “apart from urine NGAL monomer, and none of them correlated with urine light chains..” Line 250-251).

Reviewer 2 Report

Karolina Woziwodzka et al. uncover IGFBP-7 and NGAL as tubular damage markers in MM.

Points to be addressed:

1.This study design is not perfectly clear: retrospective cohort study? this should be clarified and the conclusions slightly tuned down in light of the need of statistically powered study.

2.Did the authors have the chance to also correct to revised(R)-ISS besides ISS? Indeed, a modern standpoint of MM (and CoMMpass study makes no exception), identifies "multiple" multiple myeloma, namely, the t(11;14) Myeloma, the t(4;14) Myeloma the MAF Myeloma, the Trisomic Myeloma. And if in the next ten years we will define the natural history, the prognostic features, the mechanisms of transformation, the specific strategies to overcome each one of these diseases this will be a great result to achieve. This holds particularly true for EMD and PCL. Did the authors have any clue regarding this biological background in their investigations and/or across the interrogated dataset?

3.lipocalin-2, also known as PMN gelatinase-associated lipocalin (NGAL), is produced and released by PMNs and the epithelium and generally speaking, by the MM niche. It is further known that While direct adhesive interactions between PMN and many TJ proteins have not yet been shown, it is now well established that epithelial inflammation and PMN transepithelial migration are associated with alterations in the expression levels of several TJ proteins which, in turn, have pronounced effects on the barrier and homeostasis. Proinflammatory cytokines such as interferon-gamma (IFN-γ) and tumor necrosis factor-alpha (TNF-α) that are expressed and released abundantly during IBD, have been shown to have dramatic effects on the internalization of TJ proteins such as occludin, JAM-A, and claudins. Among those, direct contact of endothelial cells with multiple myeloma cells would enhance JAM-A levels. Then it got even more interesting, as the cell adhesion molecule JAM-A has remarkable features: it can interact with itself if expressed on two opposing cell types. Furthermore, if JAM-A is shed by a cell, the soluble form of the JAM-A molecule can bind to cell-bound JAM-A, which in turn even enhances its binding capacity. What ensues is a vicious cycle of malignant plasma cells expressing and shedding JAM-A, increasing JAM-A expression on endothelial cells and stimulating blood vessel formation. In turn, increasing numbers of JAM-A-overexpressing endothelial cells can now better bind malignant plasma cells, which now find more interaction partners and by increasing the multiple myeloma niche space can produce more JAM-A.  In light of the authors' findings, and the fact that insulin-like growth factor binding protein-7 (IGFBP-7), also known as tumor adhesion factor and Vasomodulin, is a secreted protein and one of multiple IGFBP-related proteins I would suggest referring to PMID: 32354870 and expand.

4. These points hold the potential to boost the interest for a general readership in the general oncology field.

Author Response

Response to Reviewer 2 

Thank you for the interest in our manuscript. We have thoroughly revised the text following the suggestions of both Reviewers. Below, we present the detailed answers to the Reviewer’s comments and the description of the modifications introduced upon revision of the manuscript.

Karolina Woziwodzka et al. uncover IGFBP-7 and NGAL as tubular damage markers in MM.

Points to be addressed:

1.This study design is not perfectly clear: retrospective cohort study? this should be clarified and the conclusions slightly tuned down in light of the need of statistically powered study.

Response: The patients were recruited prospectively and then observed for up to 27 months. We have added the explanation in Methods (paragraph 2.1, lines 100-102 and 111-113). The first version of the manuscript concentrated on the associations between the measured markers of renal tubular injury and the renal function at the start of the study. However, following the request of another Reviewer, we have added the information about mortality and the changes in kidney function during the follow-up as a last paragraph of Results (paragraph 3.4, lines 277-293).

2.Did the authors have the chance to also correct to revised(R)-ISS besides ISS? Indeed, a modern standpoint of MM (and CoMMpass study makes no exception), identifies "multiple" multiple myeloma, namely, the t(11;14) Myeloma, the t(4;14) Myeloma the MAF Myeloma, the Trisomic Myeloma. And if in the next ten years we will define the natural history, the prognostic features, the mechanisms of transformation, the specific strategies to overcome each one of these diseases this will be a great result to achieve. This holds particularly true for EMD and PCL. Did the authors have any clue regarding this biological background in their investigations and/or across the interrogated dataset?

Response: We decided to explain this fact and expand the limitations of our study:” Moreover, in this study we used Multiple Myeloma International Staging System (ISS) instead of Revised Multiple Myeloma International Staging System (R-ISS) due to limited results of chromosomal abnormalities performed by iFISH (15 patients out of the group of 124). The accessibility to proper treatment is limited, therefore a part of patients are administered drug programs, which are not in line with mSMART tool (Mayo Stratification for Myeloma and Risk-Adapted Therapy) and do not include the need of defining the genetic and biologic features of MM.” (lines 394-400)

3.lipocalin-2, also known as PMN gelatinase-associated lipocalin (NGAL), is produced and released by PMNs and the epithelium and generally speaking, by the MM niche. It is further known that While direct adhesive interactions between PMN and many TJ proteins have not yet been shown, it is now well established that epithelial inflammation and PMN transepithelial migration are associated with alterations in the expression levels of several TJ proteins which, in turn, have pronounced effects on the barrier and homeostasis. Proinflammatory cytokines such as interferon-gamma (IFN-γ) and tumor necrosis factor-alpha (TNF-α) that are expressed and released abundantly during IBD, have been shown to have dramatic effects on the internalization of TJ proteins such as occludin, JAM-A, and claudins. Among those, direct contact of endothelial cells with multiple myeloma cells would enhance JAM-A levels. Then it got even more interesting, as the cell adhesion molecule JAM-A has remarkable features: it can interact with itself if expressed on two opposing cell types. Furthermore, if JAM-A is shed by a cell, the soluble form of the JAM-A molecule can bind to cell-bound JAM-A, which in turn even enhances its binding capacity. What ensues is a vicious cycle of malignant plasma cells expressing and shedding JAM-A, increasing JAM-A expression on endothelial cells and stimulating blood vessel formation. In turn, increasing numbers of JAM-A-overexpressing endothelial cells can now better bind malignant plasma cells, which now find more interaction partners and by increasing the multiple myeloma niche space can produce more JAM-A.  In light of the authors' findings, and the fact that insulin-like growth factor binding protein-7 (IGFBP-7), also known as tumor adhesion factor and Vasomodulin, is a secreted protein and one of multiple IGFBP-related proteins I would suggest referring to PMID: 32354870 and expand.

Response: Indeed, both tubular injury markers (NGAL and IGFBP-7) have still unexplained fully, common fields with pathogenesis of MM development. Using monomer NGAL in our study was a trial to emphasize its tubular source, that we highlighted: “NGAL is produced and released during granulocyte maturation in bone marrow by polymorphonuclear myeloid-derived suppressor cells (PMN-MDSCs). The dimer form is the major molecular form of free NGAL secreted by neutrophils and its overexpression is observed in blood cells from patients with all types of leukemia (…) In an effort to reduce the interreference of neutrophil production, our assay was based on the monomeric form of NGAL (that is specifically associated with tubular epithelial stress), rather than dimeric molecule, with the renal tubular epithelium as the purported site of origin.” (lines 332-335 and 338-340) 

According to Reviewer’s suggestions we profound the interactions of tubular injury markers with their protentional influence on MM development and pivotal role in (1) MGUS to MM transformation and MM progression that has a strong implication on future new treatment lines and may also interfere with blocking the chronic kidney disease progression or remaining it stable. We decide to extend discussion and emphasize this fact: “Noteworthy, NGAL in different hematologic malignancies was associated with vascular endothelial growth factor (VEGF), and soluble receptor for advanced glycation end-products (sRAGE) in bone marrow [z]. Moreover, angiogenic switching is a key process during transition from premalignant asymptomatic MGUS to developed MM and is considered to predict MM progression. One of stimulating MM-associated angiogenesis proteins is the junctional adhesion molecule-A (JAM-A) a potential target for future MM therapy [a]. Similarly, IGFBPs family in the bone marrow microenvironment plays important role in progression and treatment resistance in MM, though extracellular IGF system in MGUS and MM did not include IGFBP-7 [b].” (lines 352-359)

  1. These points hold the potential to boost the interest for a general readership in the general oncology field.

Response: Based on foregoing rephrases concerning molecular MM pathogenesis and development and their common field with renal damage pathogenesis we trust to reinforce oncologist interest in our study.

Round 2

Reviewer 1 Report

I do not have any more comments for this article because I considered that this revised article was suitable to publication in "Medicina".

Reviewer 2 Report

The authors have clarified several of the questions I raised in my previous review. Most of the major problems have been addressed by this revision.